

# Using ground-penetrating radar, topography and classification of vegetation to model the sediment and active layer thickness in a periglacial lake catchment, Western Greenland

Johannes Petrone[1], Gustav Sohlenius[2], Emma Johansson[1,5], Tobias Lindborg[1,3], Jens-Ove Näslund[1,5], Mårten Strömgren[4]

[1]Swedish Nuclear Fuel and Waste Management Company, Box 250, SE-101 24, Stockholm, Sweden
[2]Geological Survey of Sweden, Box 670, SE-751 28, Uppsala, Sweden
[3]Department of Forest Ecology and Management, Swedish University of Agricultural Science, SE-90183 Umeå, Sweden
[4]Department of Ecology and Environmental Science, Umeå University, SE-901 87 Umeå, Sweden
[5]Department of Physical Geography, Stockholm university, SE-106 91 Stockholm, Sweden

*Correspondence to*: Johannes Petrone (johannes.petrone@skb.se)

**Abstract.** The geometries of a catchment constitute the basis for distributed physically based numerical modeling of different geoscientific disciplines. In this paper results from ground-penetrating radar (GPR) measurements, in terms of a 3D model of total sediment thickness and active layer thickness in a periglacial catchment in western Greenland, are presented. Using the topography, thickness and distribution of sediments are calculated. Vegetation classification and GPR measurements are used to scale active layer thickness from local measurements to catchment scale models. Annual maximum active layer thickness varies from 0.3 m in wetlands to 2.0 m in barren areas and areas of exposed bedrock. Maximum sediment thickness is estimated to be 12.3 m in the major valleys of the catchment. A method to correlate surface vegetation with active layer thickness is also presented. By using relatively simple methods, such as probing and vegetation classification, it is possible to upscale local point measurements to catchment scale models, in areas where the upper subsurface is relatively homogenous. The resulting spatial model of active layer thickness can be used in combination with the sediment model as a geometrical input to further studies of subsurface mass-transport and hydrological flow paths in the periglacial catchment through numerical modelling. The data set is available for all users via the PANGAEA database, https://doi.pangaea.de/10.1594/PANGAEA.845258

## 1 Introduction

Recent climate warming, and its effect on near-surface hydrothermal regimes, is predicted to have a pronounced effect in northern latitudes (e.g. Maxwell and Barrie, 1989; Roots, 1989; MacCraken et al., 1990; Burrows et al., 2011, Larsen et al., 2014). In permafrost regions, where parts of the subsurface stay permanently frozen, one noticeable effect of global warming is a thickening of the active layer (Larsen et al., 2014). The active layer constitutes an upper sediment and/or bedrock layer that undergo annual freezing and thawing. Thickening of the active layer has biogeochemical, hydrological and



geomorphological consequences due to the exchange of energy in the form of moisture and gases between the terrestrial and atmospheric system, which occurs in the thawed subsurface (Nelson et al., 1993; Weller et al., 1995). Additionally, a long-term thawing active layer may influence the carbon cycling in the landscape; old carbon stored in the permafrost may be released implying a positive feedback to climate warming (Walter et al., 2006, Shuur et al., 2015).

Dramatic changes in the Arctic during the last decades have, among other things, lead to an increase in the length of the melting season and changes in precipitation patterns (Macdonald et al., 2005). To understand future local and regional effects of a warming climate in permafrost regions, it is first necessary to understand the present-day conditions and processes. Modeling the active layer and its variation, as well as any underlying sediments, is needed to account for different scenarios in which the active layer will vary in depth when surface conditions are changing. In addition, information

regarding the properties and spatial distribution of the sediments and the active layer constitute valuable information when studying the hydrology and mass-transport in permafrost regions.

Conventional methods to identify permafrost features or bedrock under Quaternary deposits in permafrost regions, e.g. through boreholes or excavations, can be problematic and time-consuming due to the frozen conditions. In addition, such methods provide limited information on the spatial distribution of such features. Methods for investigating the spatial

variability of the active layer includes e.g. airborne electromagnetic measurements (AEM) (e.g. Pastick et al., 2013), spaceborne interferometric synthetic aperture radar (InSAR) (e.g. Liu et al., 2012) and ground penetrating radar (GPR) (e.g. Wu et al., 2009; Stevens et al., 2009). In this study we use GPR to investigate shallow sub-surface features, such as sediment thickness and the depth of the active layer. GPR surveying has been commonly used since the mid-1990s in geophysical studies that requires high vertical resolution with little or no disturbance to the investigated area (Neal, 2004). The method is

also very flexible, as real-time acquisition and interpretation of the data immediately gives preliminary results. The suitability of the GPR method to map subsurface features in periglacial environments is due to the significantly different electrical properties between ice, water, sediment, bedrock and air. Relative dielectric permittivity is commonly used to determine the wave velocity of electromagnetic waves transmitted and received from the GPR. The dielectric permittivity is highly influenced by water content (Neal, 2004). It is therefore an effective method to map the permafrost boundary during

summer as well as mapping the subsurface bedrock (i.e. sediment thickness) when the sediments are frozen. Many studies using GPR in permafrost environments have been carried out (e.g. Wu et al., 2009; Stevens et al., 2009) as well as using the method to investigate the properties of the active layer (e.g. Ermakov and Starovoitov, 2010; Jørgensen and Andreasen, 2007; Doolitle et al., 1990; Gacitúa et al., 2012). Gacitúa et al. (2012) also showed that the difference in vegetation cover is closely tied to varying moisture content of sediments in periglacial environments.

The studies listed above, however, only focus on single and local profile investigations. The purpose of this study is to draw conclusions about subsurface conditions over larger areas, such as individual lake catchments. Here we use GPR to gather information about the relative dielectric permittivity (i.e. water content) of the sediments above the permafrost. The water content of the soil influences the composition of vegetation and the thickness of the active layer. A correlation between active layer thickness (ALT) and vegetation can therefore be expected. This correlation is used to construct a model of the



spatial distribution and variation of the active layer and upscale results from concentrated measurements to a catchment-wide model.

In the present study we use GPR during different seasons to measure the depth to bedrock and ALT within a small lake catchment, in this paper referred to as Two Boat Lake (TBL), situated close to the Greenland ice sheet (Fig. 1). The sediment thickness within the catchment has been correlated to topography, i.e. the thickest sediment layers are found in flat areas of the valley floors (Petrone, 2013). This established relationship, in combination with field surveys and sampling, forms the basis for the construction of a model of sediment thickness and general stratigraphy. For the active layer we analyse the results in combination with probe measurements to estimate permittivity of the underlying sediments and remote sensing to classify the vegetation. The relationship between vegetation and permittivity (Gacitúa et al., 2012) is used to calculate active layer depth for all GPR measurements. The result is a 3D model of sediment thicknesses, sediment types and maximum thicknesses of the active layer, on catchment scale (~1-2 km2). Data on meteorological and hydrological conditions and properties of the TBL catchment was published by Johansson et al. (2015), and Lindborg et al. (2016) published data on biogeochemistry from TBL. This new data on ALT and sediment thickness constitute valuable complementary input data when setting up distributed physically based hydrological and biogeochemical numerical models of the catchment. Since the active layer model is superimposed on the sediment model, it is also possible to study effects of a changing climate by varying the active layer depth. An increased depth of the active layer activates new hydrological flow paths above the present permafrost. The developed 3D model presented in this study may constitute a geometrical input to other models studying the effect of those processes. By correlating different types of vegetation classes found in periglacial environments to subsurface wetness conditions, we construct models extending outside of the initial point measurements.

## 2 Methods

Field investigations were carried out between the years 2010 and 2013. The methodology was conducted stepwise and each step is summarized in Fig. 2. The methods can be divided into two categories/models; one producing a catchment-wide coverage of the maximum active layer thickness and one focused on the stratigraphy and thickness of the sediments. All field investigations were carried out in the catchment of TBL, a general site description is provided in section 2.1.1 below followed of a more detailed description of the Quaternary geology of the catchment in 2.1.2. The investigations techniques and modeling methodologies for the DEM, the thickness of the active layer and the sediment thickness are described in section 2.2-2.4. All data presented in the present paper, associated measurement techniques and in which model the data is used, are listed in Table 1.



## 2.1 Site description

### 2.1.1 General site description

The TBL catchment (described in Johansson et al., 2015), is situated in close proximity to the Greenland Ice Sheet and approximately 30 km from the settlement of Kangerlussuaq, western Greenland (Fig. 1). The Kangerlussuaq region,
reaching from the coast towards the ice sheet contains an extensive (>150 km wide) ice free area, dominated by an undulating periglacial tundra environment with numerous lakes. The annual corrected precipitation (P) recorded by the local automatic weather station (AWS) in the TBL catchment, installed in April 2011, was 365 mm in 2012 and 269 mm in 2013 (Johansson et al., 2015). This is approximately twice as much as the precipitation measured in Kangerlussuaq for the corresponding periods (Cappelen, 2014). The mean annual air temperature (MAAT) at TBL is -4.3°C for the same period
(Johansson et al., 2015) and -4.8°C in Kangerlussuaq (Cappelen, 2014).

The cold climate has resulted in a region of continuous permafrost. The permafrost is, in turn, interrupted by through taliks, i.e. local areas with no permafrost, often situated below larger lakes (Christiansen and Humlum, 2000). A bedrock borehole below TBL, and the results from the investigations made in the instrumented drill hole, have shown that the lake is underlain by a through talik and that the permafrost in surrounding land areas reaches a depth of around 300 to 400 m (Harper et al.,
2011). The active layer freezes and thaws in cycles directly related to the seasonal variations in air temperature, with a maximum thawed depth at the end of summer.

The area of TBL is 0.37 km2 and its catchment covers an area of 1.56 km2 (Johansson et al., 2015). There is a height difference of around 200 m from the deepest part of the lake (~30 m water depth) to the highest point in the south western part of the catchment (Fig. 3).
Most of the catchment has a sparse but continuous vegetation cover, with some exceptions of areas with exposed bedrock or till. Vegetation is in general dominated by dwarf-shrub heath. There are no trees, and the shrubs rarely exceed 0.5 m in height. There is a fairly limited amount of vascular plant species, but the spatial distribution and density of individual species is highly variable in different parts of the catchment. The vegetation has been classified into several categories (Fig. 3A), based on field surveys and classification in aerial photographs (Clarhäll, 2011). Heath covers a large portion of the
catchment; major plants in this class are *Betula nana, Salix glacua* and *Vaccinium uliginosum*. The vegetation is classified as either *Betula* or *Salix* if any one plant dominates the coverage. In general, the class Heath constitutes a mixture of plant species where dwarf shrubs are a significant constituent. Barren surfaces (Fig. 3A), completely lacking vegetation, are present in exposed areas where wind has eroded the silt layer and exposed underlying till or bedrock. Grasslands on ridges and exposed dry areas (Fig. 3A) are also scattered around the catchment. The catchment in general constitutes a very dry
environment, but a few topographically low areas can be saturated with water during and after events of high precipitation, as well as when the snow pack melts during spring time. Areas with relative high soil water content have thus developed and they are here termed wetlands.



### 2.1.2 Quaternary geology of the study area

The geographical distribution of sediment types was determined during field surveys in the summers of 2010 and 2011. Sediment samples were taken from seven locations (Fig. 3) and analysed for grain-size distribution. Results from the field observations were used to construct a conceptual model illustrating the stratigraphical distribution of the most commonly occurring sediment types. The results are partly reported by Clarhäll (2011).

The Quaternary geology within the catchment was first described by Clarhäll (2011). An updated sediment map, based on new results from later excavations and field observations, are presented in the present paper (Fig. 3B). The uppermost sediments are dominated by eolian silt, underlain by till, which is also representative of the regional area (Clarhäll, 2011). Deposition of the eolian silt has occurred since at least ca. 4,750 years BP (Willemse et al., 2003), when the ice margin was situated further inland (van Tatenhove et al., 1996). Surficial glacial till and bedrock outcrops can be found in places where erosion has removed the eolian silt. The boundary is often sharp with a scarred surface revealing the underlying till. The till is loosely compacted and is dominated by sand and gravel with low content of silt and clay, which indicate a marginal deposition in front of the ice sheet (Clarhäll, 2011). Glaciofluvial deposits are also found in several areas, mainly in the northern regions of the catchment. Stratigraphical studies have shown that the till contains layers of water laid deposits and the hydrological properties of the till and the glaciofluvial deposits can therefore be regarded as comparable. In the central and low-laying parts of the major valleys, accumulation of organic material and silt deposition have resulted in areas of peaty silt. The floor of the lake is to a large part covered by currently accumulating silt. Permafrost related processes have also lead to the development of ice-wedge polygons in local areas where the sediments are characterised by a relatively high water content.

### 2.2 Digital elevation model

A first digital elevation model (DEM) for TBL and its catchment was constructed in 2011 (Clarhäll, 2011). In 2012, a LiDAR survey and a bathymetric survey of the TBL catchment area were carried out. A new improved DEM, with the same extent as the previous one, was constructed within the present study from the newly aquired data.

The LiDAR data was collected for a large part of the TBL catchment using a Riegl Z-450 (2,000 m range) laser scanner. Survey stations and Ground Control Points (GCP) were surveyed and georeferenced to geodetic control quality (~cm precision) using a Leica 1200 GPS receiver processed by Leica Geo Office and corrected to a local GNSS (Global Navigation Satellite System) base station located with a baseline distance of less than 5 km. The LiDAR data was projected to UTM 22N, adjusted to a local GR96 (Greenland Reference System 1996) datum point in Kangerlussuaq and converted





from WGS84 ellipsoid heights to the local elevation datum. The vertical offset between the ellipsoid height and the local datum is – 31.1 m.

Approximately 21 million elevation values were collected during the LiDAR data measurement. These values were recalculated to mean values within 1 m cells. In the terrestrial areas not covered by LiDAR data, elevation data from the previous DEM (Clarhäll, 2011) was used. LiDAR data within 2 m from the lake shoreline was removed to obtain a smoother transition between the lake and the terrestrial area in the DEM.

The bathymetric survey of TBL was made with a combined echo sounder and GPS (Humminbird 798ci HD SI) attached to a boat. Positions, water depth and bottom hardness were stored every second during the measurements. Incorrect measurements were subsequently removed and in some areas depth data were thinned out to obtain a similar data density over the surveyed area. After these corrections, approximately 18,500 measurements remained. A monitoring of lake level variations was initiated in September 2010 (Johansson et al., 2015). All water depth values from TBL were adjusted to the lake level at the beginning of the monitoring and were converted to the coordinate system used for the LiDAR data. Points were added along the lake shoreline every 5 m using the extension of the lake shown in Clarhäll (2011). The lake shoreline was adjusted to the LiDAR data and a lake level of 336.4 m a.s.l. was determined. The depth data from TBL was recalculated from the lake level, i.e. from depth in metres below 336.4 m a.s.l.

Finally, terrestrial data and depth values were merged to a data set of approximately 1.3 million points. From this data set, a 5 by 5 m resolution DEM was constructed. The interpolation of point values was done using the Geostatistical Analysis extension in ArcGis 10.1. Ordinary Kriging was chosen as the interpolation method (Davis, 1986; Isaaks and Srivastava, 1989). The resulting DEM is available at PANGAEA (DOI:10.1594/PANGAEA.845258 ).

**2.3 Thickness surveys of sediments and active layer**

The total sediment thickness in the catchment was investigated by GPR in April 2011 (Petrone, 2013). A MALÅ X3M shielded antenna GPR system, with a central frequency of 250 MHz, was used. A dGPS was connected to the GPR to log the position of each measurement. Each profile was measured by dragging the GPR equipment at constant walking speed. The measurements were carried out in straight profiles perpendicular to each other to analyze the spatial variation in sediment thickness. The methodology is thoroughly described in Petrone (2013).

To establish absolute sediment depth values, the start and end points of each transect was situated at or nearby bedrock outcrops. This allowed the bedrock reflector to be easily identified and continuously traced along the transects. The relationship between sediment thickness and topography (DEM) constitute the basis for the sediment model. The updated DEM, described in section 2.3, was used to extract the steepness of the topography. The surface sediment map (Fig. 3B) was also used as input to provide the spatial distribution of sediment types.



The thickness of the active layer was investigated by probing in 83 points in August 2011 using a steel rod (Fig. 3B). All points were located in transects at semi-regular intervals covering different soil types, vegetation and topographic properties (such as elevation, slope and aspect). An additional GPR survey was carried out simultaneously to the probe measurements, using identical GPR equipment as during the earlier measurements. This survey was concentrated to three sub areas (Fig. 3).

All GPR profiles related to the active layer surveys were processed and analyzed in the same way as in Petrone (2013). Low and high frequency noise was reduced by applying dewow and bandpass filters and each profile was geo-referenced. A clear reflector, identified as the reflection of the boundary between thawed and frozen sediments, can be identified in most profiles. In profiles with a less clear and/or discontinuous reflector, or in cases of extremely noisy radargrams, the data was discarded. All remaining selected points, representing the interpreted travel time from the surface to the permafrost boundary

and back, are seen in Fig. 3A.

## 2.4 Modeling of active layer depth and sediment thickness

### 2.4.1 Vegetation class and radar wave velocity correlation

As previously mentioned, recent studies involving GPR and the active layer have shown that there is a strong influence on

water content, and thus electromagnetic wave velocity, from different vegetation types (Gacitúa et al., 2012). A similar approach is used in the present study. In order to calculate the depth of the active layer it is necessary to know the wave velocity in the medium. Eolian silt covers a major part of the catchment, and it is considered homogeneous down to the underlying till. Thus, mainly water content and organic matter varies in the sediment layer, hosting the active layer. The variation in vegetation within the study area can therefor be correlated to soil water content (and hence wave velocity in the

eolian silt).

Probe transect 1 is situated in the northern valley of the catchment. The results and interpretation from the transect is shown in Figure 4. Processed GPR data is displayed in Fig. 4A, showing travel time to the interpreted permafrost boundary along the transect. The identified permafrost table depth and the probed transect depths are presented in Figure 4B. Electromagnetic wave velocity was calculated for the probed locations, shown under the probe bars. The corresponding

ground surface altitude and vegetation class variation is shown in Fig. 4C, along with any permafrost features. Mean values of radar wave velocity were assigned to corresponding vegetation classes (heath: 0.059 m/ns, wetland: 0.045 m/ns, *betula*: 0.058 m/ns). Grassland, absent along transect 1, was assigned a velocity value based on heath and *betula* (0.0585 m/ns), assumed to be similar in soil water content. The wave velocity near the shore line (water class) was assigned a velocity of 0.05 m/ns based on saturated silt (Neal, 2004). Assigned velocities and the source for the estimation are presented in Table 2.




### 2.4.2 Upscaling of active layer depth to catchment scale based on vegetation classes

The vegetation map presented in Clarhäll (2011), shown in Fig. 3A, was based on areal photographs of the catchment. However, the classification was not of satisfactory detail for the analysis in the present study since it had to better match the high resolution of the GPR measurements. In order to increase the spatial resolution of the vegetation map, the same aerial

photographs were used to define vegetation classes within the areas investigated by GPR and probe measurements (Fig. 3A). To this end, a so-called supervised classification on pixel basis was performed using ArcGIS. Based on pre-defined areas, with assigned classes (training areas), the software analyzes and matches groups of pixels and assigns every pixel to the predefined classes. The same vegetation classes were used for both the high and low resolution vegetation maps. The resulting maps provided high resolution information on vegetation cover over the areas investigated by the GPR and probe

measurements.

All reflectors (Fig. 3), i.e. points within the area with a known travel time for the electromagnetic wave to the permafrost table, were assigned a wave velocity based on Table 1 and the vegetation in that point. The velocity estimations were checked in a separate probe transect 2 (2 in Fig. 3A). The processed GPR data, plotted topography and vegetation, as well as the permafrost boundary and probe measurements of transect 2, are shown in Fig. 4 D-F. The comparison between measured

and modeled values of the active layer thickness (Fig. 4F) shows a good accuracy ($R2 = 0.84$) when using the estimated velocities in Table 2.

To calculate the active layer thickness for all reflector points in Fig. 3, the re-classified vegetation maps described above were used. Each GPR measurement was assigned the estimated velocity from Table 1, corresponding to the vegetation class. The results are summarized in Table 3, listing the different vegetation classes and the calculated active layer depths. The

mean active layer thickness for each vegetation class was used to construct the catchment model of the active layer. Since no measurements of active layer thickness were made over bedrock, the value of 2.0 m for the depth corresponding to the barren class has been taken from Harper et al. (2011) which is based on bedrock borehole temperatures in the area. The mean active layer thickness values of each vegetation class were fed into the lower resolution map of the vegetation coverage, resulting in the catchment-wide model of the variation in active layer thickness. Sharp boundaries between vegetation classes (and thus

active layer depth) were smoothed using a mean filter.

As an additional source of information regarding the active layer depth, data from a soil temperature station (Fig. 3B), continuously monitoring temperatures every 3 hours in the sediments at 0.25, 0.5, 0.75. 1.0, 1.5 and 2.0 m depth below surface was used (Johansson et al 2015). The station is located in heath vegetation along transect 2 (Fig. 3B). The data has been used partly to verify the result of the active layer model, as supportive information regarding the seasonal evolution of

the active layer within the catchment and to analyze annual differences in the thaw depth of the layer.

## 3 Results

### 3.1    Sediment thickness

The final result is a 3D model illustrating the thickness of the most commonly occuring sediment types. Based on hydrological properties, the sediments have been divided into three major classes; glacial deposits, eolian silt and lacustrine

silt. The classes are further described below.

*Glacial deposits.* This class includes both till and glaciofluvial deposits, since these deposits have similar hydrological properties, see above. The GPR measurements show that the thickness of the glacial deposits range from 0 m where bedrock is exposed, to 10 m in the central and low laying parts of the valleys and the lake floor (Petrone, 2013). The slope of the ground surface has been assumed to mainly dictate the thickness of this layer. Based on results from the GPR measurements

(Petrone 2013) and the sediment map, larger flat zones (slope < 5°) has been assigned a thickness of 10 m while steeper (slope > 60°) sections lacks any glacial deposits. A linear interpolation was used to assign the  thickness value between  the minimum and maximum values. In addition, the modelled sediment thickness was manually increased where the glacial deposits comprise positive morphological landforms, such as along ridges.

*Eolian silt.* This sediment class covers a major part of the catchment and is superimposed on the glacial deposits. In steep

topography, no silt has accumulated or has been eroded and re-deposited elsewhere. In several flat and exposed areas, wind has eroded and removed any vegetation and silt, revealing the surrounding silt and underlying till. The thickness of the silt in these regions is approximately 0.2 m. The maximum depth of the silt is found in the central and low-laying parts of the valleys, reaching a thickness of 1 m. Thus, the thickness of the eolian silt is also largely dependent on the topography, as well as the distance from any eroded sections. From the field observations and coverage maps, aoelian silt has been assigned

a thickness of 0.2 m in proximity to eroded sections and steeper parts (slope > 50°) to 1.0 m in the flat (slope < 5°) and low-laying sections of the catchment (Clarhäll, 2011).

*Lacustrine silt*. These sediments are superimposed on the glacial deposits at the lake floor. Due to wave action, no deposits of lacustrine silt are found in the shallow (< 2 m) parts of the lake. In the deeper parts, the deposits has a thickness of around 1.5 m (Petrone, 2013). The thickness of the lacustrine silt deposit between the shallow and the deeper part are based on linear

interpolation.



Fig. 5 shows the final model of the thickness of each sediment layer. Glacial deposits (Fig. 5A), mainly till, have been deposited on top of the bedrock during periods with a more extensive Greenland ice sheet. At higher elevations, and where the slope is steep, no glacial deposits are generated in the model. Thinner layers of sediment are found in close proximity to exposed bedrock. In flatter terrain and in central parts of valleys, the sediment thickness increases to around 10 m. A thin
layer of eolian silt covers much of the glacial deposits (Fig. 5B). These eolian deposits have a maximum depth of 1 m in flat terrain and generally decrease where the topography steepens. Fig. 5B also shows the lacustrine silt at the lake floor. Lake deposits are only found at depths below 2 m in the model, and they gradually increase with depth, reaching a maximum of 1.5 m at the deepest part of the lake.

The results have been used to construct a schematic sediment distribution model within the catchment (Fig. 6). Till makes up
the majority of the volume of sediment and is present in all areas of the catchment. At the valley floors , the eolian silt is partly rich in organic material due to higher soil water content, and is denoted peaty silt. The maximum thickness of the sediments can be found where the glacial deposits constitute positive landforms, such as along glaciofluvial ridges.

### 3.2 Active Layer

Based on the methodology described in section 3.2 and 3.3, a catchment-wide model of the active layer was produced (Fig. 7A). The shallowest active layer is found within the valley wetlands, often in close proximity to the lake and has an average maximum thickness of 0.48 m. Areas dominated by heath, betula and grassland, generally constituting drier areas, have a thicker maximum active layer (between 0.68 m and 0.80 m). The thickest active layer is however found in the barren areas with exposed bedrock (2.0 m) where the pre-defined value of 2.0 m is used, see above. The active layer depth gradually
increases along the shore where it evolves into the talik feature. The results and the correlation between active layer depth and vegetation is schematically visualized in Fig. 7B. Polygonal surfaces and ponds where water is residing for longer periods of time have not been incorporated into the catchment-wide model. Probe measurements in the field has shown that these have a relative thin active layer (approximately 0.3 m) and has thus been included in the schematic model (Fig. 7).

Based on the soil temperatures, monitored in the vegetation class heath, the development of the thaw depth in the active layer
can be visualized in a contour plot which covers the entire period of 2011-2015 (Fig. 8). The maximum observed thaw depth in 2011 (Fig 8), when the GPR measurments were carried out, is 0.75 m, which is within the range of the active layer depth for the vegetation class heat calculated to 0.68 ±0.09 m (Table 3). The annual mean maximum thaw depth for the whole period is ~0.9 m, indicating the GPR measurments were carried out a in cold year, also can be seen in Figure 8. A consistent thaw depth, deeper than 25 cm, is reached in June all years.  The plot doesn't show shallow (< 25 cm) temperature



fluctuations but temperatures in the uppermost sediments rise above 0°C in late April to early May (Johansson et al., 2015). In June, the air temperature is always above freezing and the thawed depth gradually migrates downward in the soil. Maximum temperatures in the upper part of the soil occurs in early August, whereas the thawed layer continues to increase in thickness until early September, although the rate slow down during August. The upper part of the active layer freeze once

air temperatures drop below 0°C for longer periods. Large fluctuations in temperatures during the October monthts lead to periods with freezing and thawing of the upper part of the active layer at the same time as the lower boundary of the active layer migrates upwards. Data from the soil temperature station shows that parts of the sediments stay thawed until mid-October, and in warmer years, such as 2012, as far as into early November. In this way, different surface conditions not only leads to variation in the length of the active period, i.e. the time period when a thawed active layer exists, but also to

variation in the active layer thickness. The general pattern of the evolution of the active layer is similar each year; a rapid thaw in early June which stagnates in late August followed by a more complex refreeze in late September to October.

## 4 Discussion

GPR is a powerful tool that can be used to investigate several properties of the subsurface. During frozen ground conditions

it can be used to determine the depth to the bedrock and any intermediate stratigraphical horizons. In areas with relative homogenous sediments and a known Quaternary history, such as the present study area, combining the measurements with a digital elevation model and field observations can be used to construct a model of the different sediments types and its spatial variation in thickness. In this study we rely on radar profiles which transect bedrock outcrops and make it possible to trace the bedrock, and thus sediment thickness. Since we use the pronounced topography in the catchment to such a large

extent when modeling the sediment thickness, it might not be feasible to use the same methods in areas with less pronounced topography. GPR is also a suitable method to investigate the permafrost table in periglacial environments, due to the large difference in electrical properties between thawed and frozen sediments. Since soil water content influences the permittivity of the medium to a large degree, one must first estimate the permittivity of the material. Here we used aerial photographs to tie vegetation classes to a specific soil water content (and hence its dielectric properties), using probe measurements to

determine the absolute depth to the permafrost table. This allowed us to process the GPR measurements and resulted in over a thousand points for the active layer thickness over several vegetation classes. As can be seen in Fig. 4B and Table 3, the active layer thickness can vary significantly within small areas and within vegetation classes. It would be very time-consuming to do the probe measurements with such high resolution as was done with the GPR, especially when working in a large area such as the studied catchment. However, radar measurements can continuously map variations in active layer

thickness in a relative short amount of time. The result from the GPR showed distinct variations in thickness of the active layer between vegetation classes. The variation between classes was used to upscale the results to a catchment-wide model.





An uncertain factor is the active layer thickness in barren areas, often with little to no soil water content. Here we used temperature measurements from a nearby borehole, which showed the active layer extending down to 2.0 m in the bedrock (Harper et al., 2011). The significant increase in active layer thickness in the bedrock compared to sediments can be explained by the difference in thermal properties, namely thermal diffusivity. Bedrock has a higher diffusivity compared to

the sediments and transfer heat at higher rate (Janza, 1975); affecting the energy transfer from the surface and leads to deeper penetration of energy (heat) in a given active period. The same argument can be applied to the difference in the active layer thickness between different vegetation classes. Since the surface vegetation is directly connected to soil moisture content, differences in water content of the sediments will affect the diffusivity of the material. The diffusivity of water ($1.5 \ 10^{-3}$ $cm^2/sec$) compared to that of soil ($3$-$5 \ 10^{-3} \ cm^2/sec$) will lead to a quicker thickening of the active layer, and hence deeper

permafrost table during the active period. The energy from the surface has several sources including air temperature, precipitation and solar radiation that will also influence the active layer thickness. This relationship has not been investigated in the present study and variations in these parameters within the catchment are not accounted for in detail. However, investigations of the active layer depth have been carried out in different areas within the catchment and local variations are therefor assumed to be included in the final model results. In this study we focus on the spatial and temporal variation of the

active layer within the catchment. The permafrost boundary is at, or very near, its deepest point in August. However, it should be noted that the GPR measurements were carried out in 2011. This year is shown to have the shallowest observed thaw depth at the soil temperature station in the end of the active period. During the period of soil temperature measurements (Fig. 8), the temperature sensor at 1 m depth has never experienced temperatures below 0°C. The average thaw depth in the active layer for the period 2011-2015 is 0.9 m. The measurements and active layer model produced here reflect the year of

2011 and should therefor perhaps be seen as a low value for the thickness of the active layer.
Long-term variations in the thawing depth or in the duration of the active period, may affect the local hydrology and transport of matter in the catchment. An increased thaw depth of the active layer might open up new pathways for water to flow, drastically altering the subsurface hydrology of the catchment. Additionally, it is unclear whether increased thawing of the active layer will lead to dryer or wetter conditions in Arctic landscapes, which affects the cycling of carbon. The

presented models of ALT and sediment thickness can be used as input in hydrological models aiming at investigating the storage and partitioning of water in Arctic landscapes under increased thawing conditions.

## 5 Conclusion

We have used a combination of remote sensing, ground-penetrating radar, field surveys, digital elevation model analysis and

probe measurements to construct two models over the catchment; one of the sediment thickness and another of the maximum

active layer thickness. The sediment thickness ranges from 0 m in bedrock outcrops areas to more than 12 m in the central

valleys.  The active layer is mostly confined within the upper 0.7 m of the sediment (excluding the bedrock), which mostly

constitute the aeolian silt. The active layer thickness varies from 0.3 m in the wetland areas to 2 m in areas with bedrock

outcrops.

We show that by using relative simple methods, it is possible to upscale local point measurements to catchment scale

models, in areas where the upper subsurface is relatively homogenous. By identifying generalized active layer thickness

values for different vegetation classes, we were able to construct a 3D model of the active layer thickness on catchment scale

based on vegetation coverage.

All data needed to construct the soil- and active layer models, and the models themselves, are freely available in

PANGAEA: https://doi.pangaea.de/10.1594/PANGAEA.845258.  The spatial information on sediment distribution with

depth is valuable when assigning hydraulic properties to conceptual and numerical hydrological models of the catchment,

which in turn may be used to model biogeochemical transport and processes in both the limnic and terrestrial system

## 6 Authors contributions

**Johannes Petrone** prepared the manuscript with contributions from all co-authors. He performed the GPR-measurements

and the analysis of all GPR-data. He established the method for upscaling the GPR-data to represent the active layer

thickness on catchment scale by combining it to vegetation data.

**Gustav Sohlenius** carried out the site investagation related to sediment thickness and properties. He was involved in the

modeling of the sediment thickness and wrote the associated parts of the manuscript.

**Emma Johansson** (hydrology and meteorology) and **Tobias Lindborg** (ecology and chemistry) are responsible for the

GRASP field program. They were both involved in the planning of all field investigations presented in this paper and they

took active part in the writing of this paper.

**Mårten Strömgren** wrote the parts describing the construction of the Digital Elevation Model and he was involved in the

associated investigations.

**Jens-Ove Näslund** was involved in the field work and took active part in the writing of the paper.



# 7 Acknowledgement

The majority of the work was conducted as a part of the Greenland Analogue Surface Project (GRASP) funded by the Swedish Nuclear Fuel and Waste Management Company (SKB). The authors would like to thank the Greenland Analogue Project (GAP) for providing additional LiDAR data of the catchment, and Kangerlussuaq International Science Support

(KISS) for providing logistical support throughout the years

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

## 9        Tables and Figure Captions

15   **Table 1. List of equipment, accuracy and in which resulting model the data is used. DEM = Digital Elevation Model, AL = Active Layer Model, STM = Sediment Thickness Model**

| Parameter | Measurement technique | Used in model | Data origin |
|---|---|---|---|
| Elevation | LiDAR | DEM | Present paper |
| Lake Bathymetry | Echo sounding | DEM | Present paper |
| Surface Vegetation Class | Aerial classification, field mapping | AL | Clarhäll et al., 2011 |
| Surface Sediment Class | Sediment sampling, excavations, field mapping | STM | Clarhäll et al., 2011 and present paper |
| Active Layer Depth | Probing, GPR | AL | Present paper |
| Depth to bedrock | GPR | STM | Present paper |



**Table 2. Summary of the results from the velocity analysis in Fig. 5.**

| Vegetation class | Electromagnetic wave velocity [m/ns] | Source |
|---|---|---|
| Wetland | 0.045 | Fig. 4B |
| Betula | 0.058 | Fig. 4B |
| Heath | 0.059 | Fig. 4B |
| Grassland | 0.0585 | Fig. 4B |
| Shore/Water | 0.05 | Neal (2009) |

**Table 3. Summary of GPR points from Fig. 3A and wave velocities from Table 1. *Value from Harper et al. (2011), based on temperature measurements in boreholes.**

| Vegetation class | Number of points | Active layer mean depth [m] | Standard deviation [m] |
|---|---|---|---|
| Wetland | 748 | 0.48 | 0.09 |
| Betula | 57 | 0.7 | 0.13 |
| Heath | 288 | 0.68 | 0.09 |
| Grassland | 42 | 0.81 | 0.05 |
| Shore/Water | 99 | 1.27 | 0.16 |
| Barren (Bedrock)* | - | 2 | - |



**Figure 1: Map showing the location of the study site (Two Boat Lake), accessible from the nearby town of Kangerlussuaq (modified from Johansson et al., 2015).**





**Figure 2: General work flow leading up to the completed models covering both active layer thickness and regolith thickness.**

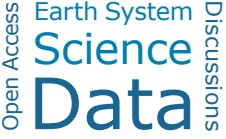

**Figure 3: (A) Vegetation map of the catchment outlined in black with defined sub areas and GPR data pick points, (B) The aerial distribution of sediment classes and bedrock outcrops within the catchment in addition to field measurements. Spatial distributions are from Clarhäll (2011).**

5    **Figure 4: Profile 1 (Fig. 3B) with A) the raw data from the GPR B) Probes active layer depths and picked permafrost reflector from the raw data. Values below probe bars represent the calculated electromagnetic wave velocity representing the points. C) Elevation, permafrost features and vegetation. Profile 2 (Fig. 3B) with D) raw data from the GPR, E) Surface altitude and vegetation and F) probe measurements and modelled values for the active layer.**





**Figure 5: Modelled sediment thickness for different classes of sediments in the catchment of Two Boat Lake. (A) Glacial deposit (till and glaciofluvial material) thickness, (B) Aeolian and lake deposits (C) Total sediment thickness including superimposed glacial, Aeolian and lake deposits.**

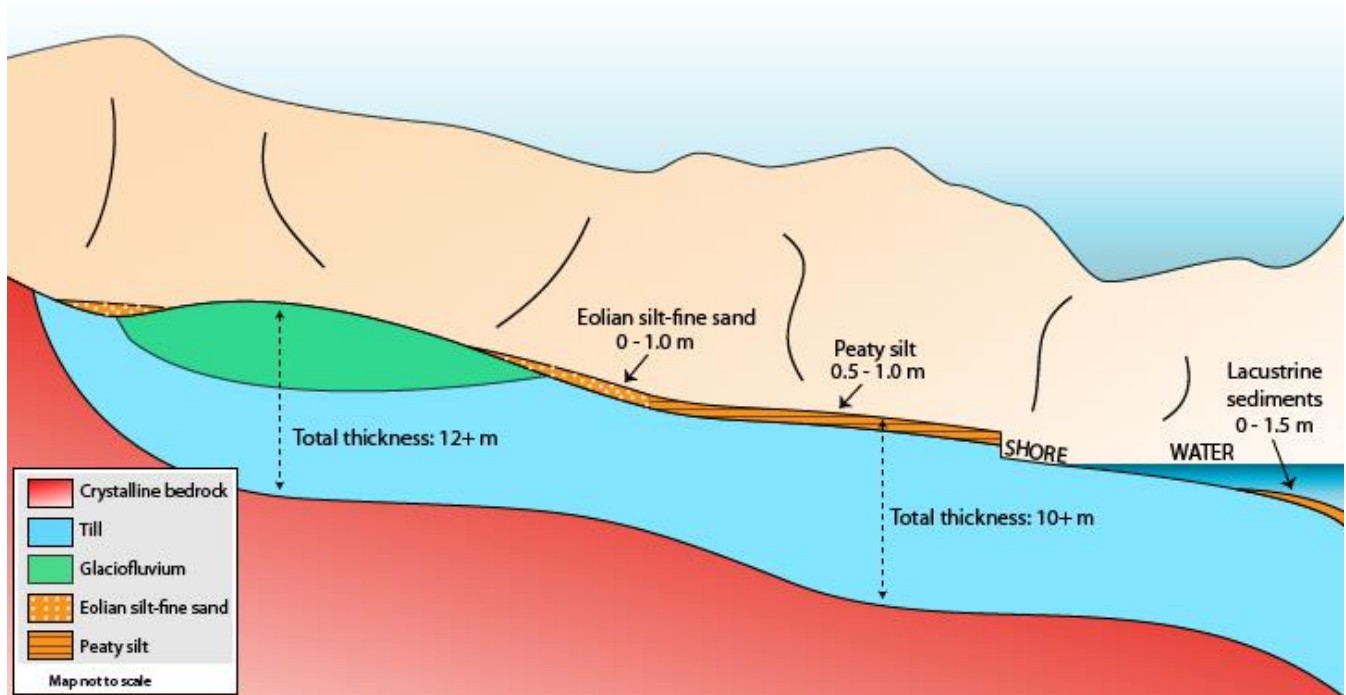

5    **Figure 6. Schematic model and presentation of the sediment thickness from the lake (right) towards higher elevations (left) in a cross-section through the valley. Sediment classes and its visual assignment are based on Fig. 3B.**






**Figure 7. Maximum active layer thickness over the entire catchment of Two Boat Lake (upper). Schematic model of the variation in active layer thickness within the catchment of Two Boat Lake and its relation to topography, regolith, permafrost features and surface vegetation (lower).**

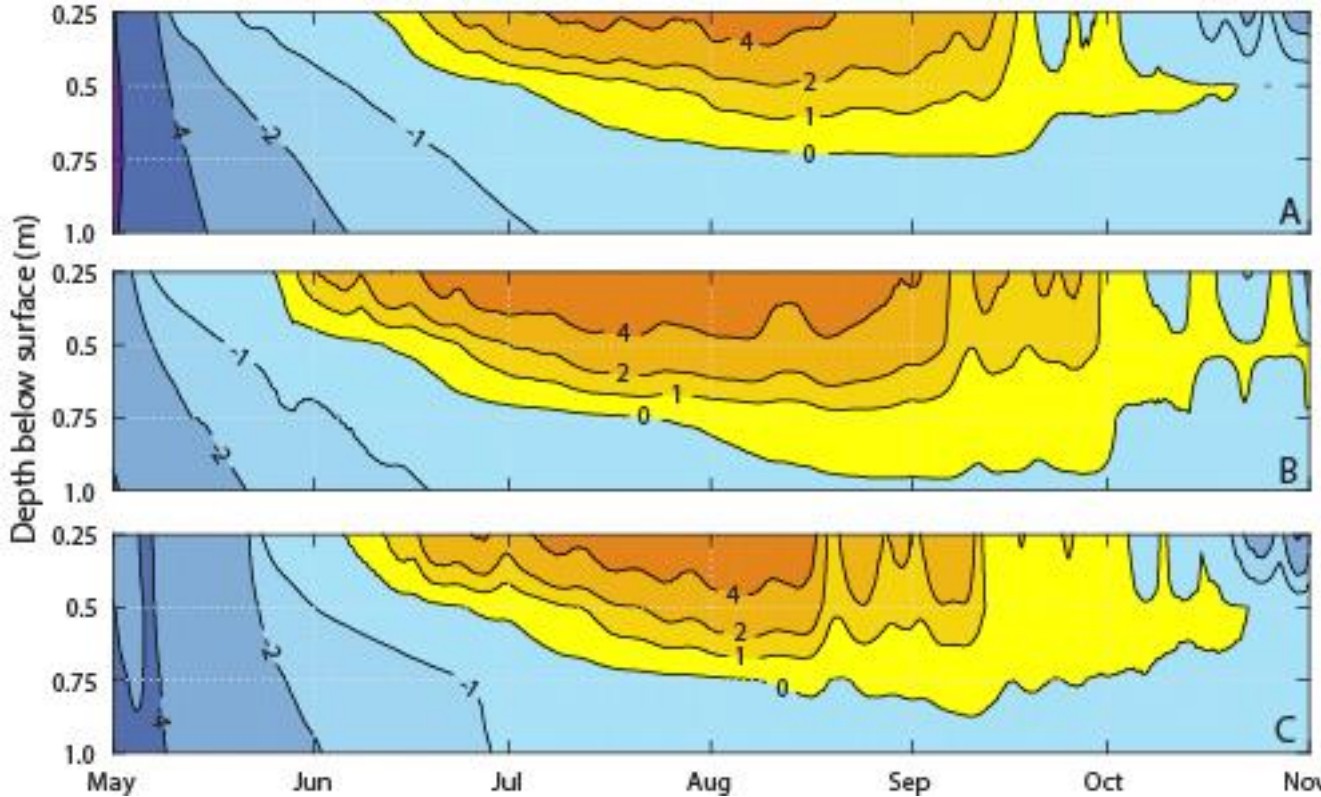

5     **Figure 8: Temperature in the upper 2 meter of sediments during the period 2011-2015. Contour values are shown in °C where yellow to orange gradients represent the active layer and blue gradients frozen soil.**