# Peer review of "Using ground-penetrating radar, topography and classification of vegetation to model the sediment and active layer thickness in a periglacial lake catchment, Western Greenland"

_Earth System Science Data, 2016_

## Referee Comment (RC1) · Johannes Petrone et al. · 22 Jul 2016

The paper is well written and presented an interesting case study. Some comments
and suggestions are presented in the following: Page 6, line 7: I suggest to improve the
description of the system used for bathymetry. In particular, I suggested to insert the
description of the GPS acquisition characteristics (e.g the acquisition has been done
with or without RTK correction) and the estimated accuracy of the obtained DTM Page
7 from line 21: I suggest to improve the description of the relation between figure 4B
and 4C. If I understood correctly, 4C is the real section of the presented GPR section.

[Figure]

If yes, I suggest to plot over figure 4C the limit of the permafrost reflector presented in figure 4B. Page 10, line 21: the indication of figure A and B in figure 7 is missing please improve accordingly. Figure 4: use the same order or presentation of section (e.g. change 4F with 4E) Figure 7: A and B are missing

---

## Referee Comment (RC2) · Anonymous Referee #2 · 2 Aug 2016

The paper describes an interesting approach to integrate environmental data (obtained from ground-penetrating radar, topography and classification of vegetation) with a model (regarding the sediment and active layer thickness) in a periglacial lake catchment. The manuscript is of overall good quality, perhaps some more detailed maps could be added in order of a more complete description of the study sites.

---

## Referee Comment (RC3) · J. Engström (Referee) · 29 Aug 2016

GENERAL COMMENTS: In general a good paper with an interesting research topic and relevant for the permafrost community. The paper addresses several issues and topics concerning this scientific field, which is good for defining the periglacial environment and processes occurring in that environment. However, a few corrections and improvements are required to clarify some of the figures and especially enhance the discussion part of the paper.

SPECIFIC COMMENTS: Fig. 4A-C and Fig. 4D-F are probably profiles 1 and 2 in Fig 3B. I presume the profile are the roughly E-W transecting profiles in the map Fig. 3B!? I would also recommend that the Fig. 4 figures should have a consistent order with GPR-profile, Probe and the model of different layers, now they are mixed between profile 1 and profile 2. Fig. 4 B and Fig. 4F why are there a difference between the profiles in the figures and what is the difference between the probe data and the modelled active layer depth from the GPR profiles? Can you please elaborate around this problem and write it out a bit more clearly.

Fig. 7 Upper part is the model of the active layer thickness in the catchment area while the lower is a schematic model, could you indicate in the upper figure where the schematic model in the lower figure is located? The text is referring to Fig. 7A and 7B but this is not indicated in the figures or the figure caption below, please adjust caption accordingly to the text.

In the Fig 8 it is missing A, B and C, please add that to the figure. Please also add A, B and C to the caption and please refer to these different sections in the text clearer. Especially section 3.2 and lines 15-20 of the discussion part would be more understandable if the reference to Fig. 8 updated accordingly to the suggestions above. The text in the discussion is unclear concerning the soil temperature measurements and especially where this measurement is performed. Can this be indicated? Where approximately is this Fig 8 for the model taken, can it be indicated on any of the maps in the earlier figures?

If these corrections and improvements to paper is done, I think that the paper will be a good contribution to the periglacial and permafrost scientific community.

---

## Author Comment (AC1) · 13 Sep 2016

Comment: The paper is well written and presented an interesting case study.

Response: We thank the Referee for taking the time to read through and assess the paper.

————————

Comment: "Page 6, line 7: I suggest to improve the description of the system used for bathymetry. In particular, I suggested to insert the description of the GPS acquisition characteristics (e.g the acquisition has been done with or without RTK correction) and the estimated accuracy of the obtained DTM..."

Response: The resulting DEM has three data sources. i) A previously refined DEM (5 meter resolution) originally based on orthophotos by Scancort. ii) Combined GPS and Echo sounding measurements in the lake. iii) LiDAR data from measurements presented in the present paper.

Details about the data processing based on information from Scancort are presented in Clarhäll et al. 2010. Two measurement campaigns with the combined GPS-Echo sounding technique were performed; one campaign in 2010 and one in 2011. Details about the measurements performed in 2010 are presented in Clarhäll et al., 2011 and the measurements performed in 2011 are described in the present paper. However, some information about the equipment is missing and therefor presented in the following. The GPS acquisition for the bathymetry was done without RTK correction, as there is no such option for this particular GPS-receiver. The combined GPS-echo sounding equipment has estimated accuracy in the range of ±1m in the horizontal plane and ±0.1m in depth.

Manuscript changes: Appendix 1 will be included in the updated manuscript with a map showing which extents of each data source used in the final interpolation of the DEM (i-iii mentioned above). In Appendix 1 details about accuracy of the different data sources and used field equipment (Humminbird 798ci HD SI) equipment are given.

——————

Comment: "Page 6, "Page 7 from line 21: I suggest to improve the description of the relation between figure 4B and 4C. If I understood correctly, 4C is the real section of the presented GPR section. If yes, I suggest to plot over figure 4C the limit of the permafrost reflector presented in figure 4B."

Response: Yes, 4C is the real section of the presented GPR section where elevation and vegetation types are illustrated. Due to different scales on the y-axes in Figure 4B and C it is hard to plot the interpreted permafrost boundary in Figure 4C. Instead, in the updated manuscript we have included information on vegetation types from 4C in Figure 4B in order to make the coupling between B and C clearer. The text in the updated manuscript is presented below.

Manuscript changes: Clarification regarding Probe transect 1, marked in Figure 3B, which is situated in the northern valley of the catchment. The results and interpretation from the transect is shown in Figure 4A-C, and the profile is presented in E-W direction. Processed GPR data is displayed in Fig. 4A, showing travel time and reflectors. The identified permafrost table reflector and active layer depths from probing are presented in Figure 4B. Electromagnetic wave velocity was calculated for the probed locations. The corresponding surface altitude and vegetation class variation is shown in Fig. 4C, along with any permafrost features. For clarity, the vegetation classes along the transect shown in Figure4C, are also included in 4B.
* * *
Comment "Page 10, line 21: the indication of figure A and B in figure 7 is missing please improve accordingly."

Response: Correct, A and B is missing.

Manuscript changes: Figure 7 is updated with A and B. The figure caption is updated accordingly.
* * *
Comment: "Figure 4: use the same order or presentation of section (e.g. change 4F with 4E) "

Response: Correct, the order should be consistent.

Manuscript changes: Figure 4 is updated and the order of the figures in D-F is now the same as in A-C.

---

## Author Comment (AC2) · 13 Sep 2016

Comment: "The manuscript is of overall good quality, perhaps some more detailed maps could be added in order of a more complete description of the study sites."

Answer: We thank the Referee for taking the time to read through and assess the paper. We believe the number of maps is sufficient. In Figure 1 the location of Kanger-lussuaq both globally and regionally is illustrated as well as the location of the site

and catchment (Two Boat Lake). Figure 3A and Figure 3B shows detailed maps of vegetation and surface soil classifications. Maps of modelled active layer depth and sediments within the catchment are also presented in the paper.

Manuscript changes: -

———————————————

---

## Author Comment (AC3) · 13 Sep 2016

Comment: Fig. 4A-C and Fig. 4D-F are probably profiles 1 and 2 in Fig 3B. I presume the profile are the roughly E-W transecting profiles in the map Fig. 3B!? I would also recommend that the Fig. 4 figures should have a consistent order with GPR-profile, Probe and the model of different layers, now they are mixed between profile 1 and profile 2. Fig. 4 B and Fig. 4F why are there a difference between the profiles in the figures and what is the difference between the probe data and the modelled active layer

depth from the GPR profiles? Can you please elaborate around this problem and write it out a bit more clearly.

Response: We realize that we should be clearer regarding the whole of Figure 4 and have addressed this partly in Author Comment 1 in the second response. To summarize: Yes, the transects are illustrated in E-W direction. The main difference between Figure 4B and Figure 4E (originally 4F) is that raw data is presented in Figure 4B to extract information which can then be applied to other transects (Figure 4E).

GPR travel time to PF boundary vs Active Layer depth at probe locations vs vegetation (Figure 4B) resulted in general wave velocities for each vegetation classification. These can be found in Table 2. We tested the velocity values using a separate transect (Figure 4E) where we used the velocities from Table 2 to model Active Layer depth from the GPR measurements. The modelled values matched the probe depths to an acceptable correlation and these wave velocities were then applied to all GPR measurements.

Changes in manuscript: The order of Figure 4A-C and 4D-F is updated so that they appear in consistent order as suggested. Both the figure caption text and the main text describing the methods have been rewritten slightly to make it clearer the connection between probe depth, GPR travel time, wave velocity, vegetation class and modelled active layer depth.
* * *
Comment: "Fig. 7 Upper part is the model of the active layer thickness in the catchment area while the lower is a schematic model, could you indicate in the upper figure where the schematic model in the lower figure is located? The text is referring to Fig. 7A and 7B but this is not indicated in the figures or the figure caption below, please adjust caption accordingly to the text."

Response: The schematic figure represents a general valley area of the catchment, ranging from the catchment boundary towards the lake.

Changes in manuscript: Black dashed lines have been added in Figure 7A to illustrate valleys found within the catchment and the figure caption has been updated to explain this.
* * *
Comment: "In the Fig 8 it is missing A, B and C, please add that to the figure. Please also add A, B and C to the caption and please refer to these different sections in the text clearer. Especially section 3.2 and lines 15-20 of the discussion part would be more understandable if the reference to Fig. 8 updated accordingly to the suggestions above. The text in the discussion is unclear concerning the soil temperature measurements and especially where this measurement is performed. Can this be indicated? Where approximately is this Fig 8 for the model taken, can it be indicated on any of the maps in the earlier figures?"

Response: A wrong version of Figure 8 has been uploaded to the manuscript. The wrong figure shows soil temperatures only for the active periods of 2011, 2012 and 2013 and not for the whole period of 2011-2015 as indicated in the text. We apologies for the inconvenience, but updating the figure to the one showing soil temperatures for the period 2011-2015 should solve the questions raised. The text in the manuscript is correct given that the figure is updated.

The location of the soil measurements is marked in Figure 3B and in Line 24 P 10 it is written that the temperature station is placed in the vegetation group heath.

Changes in manuscript: Updated to correct figure showing the whole period of 2011-2015. See attached figure.
* * *
[Figure]

**Fig. 1.** Correct figure 8.

---

## Author Comment (AC4) · 11 Oct 2016

The previous version of Figure 8 shows soil temperatures only for the periods of 2011, 2012 and 2013 and not for the whole period of 2011-2015 as indicated in the text. We apologise for the inconvenience and have updated the figure to the correct Figure 8 showing soil temperatures for the period 2011-2015. No changes to figure caption.

[Figure]

[Figure]

**Fig. 1.** Figure 8.